# AN EMPIRICAL INVESTIGATION OF GENERALIZATION DYNAMICS IN DEEP ReLU NETWORKS VIA NONLINEAR MODE DECOMPOSITION

## ABSTRACT

The ability of deep networks to generalize, effectively learning some underlying nonlinear transform from noisy data, has long been investigated. The generalization dynamics of deep linear networks have previously been solved analytically and shown to qualitatively capture some aspects of how a linear transform is learned by a nonlinear network. Here we explore zero-bias deep ReLU networks playing both roles in a teacher-student framework, where measurement of the Jacobian of the network transform with respect to every input point allows for a complete description of the transform at those points, given the piecewise-linear nature of the ReLU network transform. The singular value/mode decomposition (SVD) of the Jacobian is computed at every input point for both teacher and student networks. The evolution over training of the singular values and vectors, averaged over all inputs, provides measurements of the globally nonlinear behavior of the network transform. For a deep ReLU student network trained on data from a deep ReLU teacher network with a user-specified singular value spectrum, we show over the course of training increasing student singular value magnitudes and increasing alignments of student singular vectors with teacher singular vectors, as observed in deep linear networks. We decompose the loss over training by singular mode, and directly observe nonlinear coupling of noise to student signal singular modes as well as coupling due to competition between signal modes. Nonlinear modes are shown to occur in teacher-student scenarios for deep convolutional ReLU networks with MNIST data, where their singular vectors reveal interpretable features that the network learns at local and global levels.

## 1 INTRODUCTION

The ability of deep networks to generalize to unseen data, and thus learn an underlying input-output transform, is the primary reason for their widespread use. The source of this generalization capability, in terms of properties of the data, task, or network itself has long been investigated (Seung et al., 1992) (Saad & Solla, 1995). Using the teacher-student framework of Seung et al. (1992), Saxe et al. (2013) examined deep linear student networks trained using noisy outputs from a linear teacher network, where the teacher weight matrix had a specified singular value structure, and derived the nonlinear dynamics of the loss function during training using the singular value decomposition (SVD) (Behrmann et al., 2018; Raghu et al., 2017). Insight into successful generalization for overparameterized two-layer nonlinear networks has also been made both analytically and empirically (Advani et al., 2020; Goldt et al., 2019).

Recently, continuing in the tradition of statistical mechanics applied to neural networks (Bahri et al., 2020), there have also been developments in initialization frameworks for nonlinear networks (Pennington et al., 2017), invariants and conservation laws from symmetries in parameter space (Kunin et al., 2020; Chen et al., 2023) as well as applications of random matrix theory to generalization in trained networks (Mahoney & Martin, 2019; Martin & Mahoney, 2021; Yang et al., 2022; Pennington et al., 2018). There is also a related approach to infinite-width deep networks with the neural tangent kernel (Jacot et al., 2018; Simon et al., 2021).

Lampinen & Ganguli (2018) continued the approach of Saxe et al. (2013) and derived expressions for a deep linear network's training and generalization error over training time as a function of the network's size, the number of input examples, the task structure and the signal-to-noise ratio. The specification of the teacher transform by the singular value structure of its weight matrix allowed for a dissection of the deep linear student network via the SVD of its composite weight matrix. They found that a deep linear student network learns the input-output relationship of a linear teacher network in a "singular value detection wave"; that is, the largest singular vector from the teacher network is learned and appears first in the student network singular value spectrum, then the next largest, and so on, with each student singular value increasing in magnitude over training following a sigmoid time course. This framework is also partially applicable to deep nonlinear student networks with linear teacher networks, and they demonstrate a qualitative relationship between learning dynamics in the linear and nonlinear student cases. Saxe et al. (2013) also measured singular mode strength over training in student networks with sigmoid nonlinearities using the input-output correlation matrix for the linear teacher scenario, although this method cannot capture the dynamics induced by a nonlinear teacher network. This general methodology has also been applied to gated deep linear networks (which are globally nonlinear) (Li & Sompolinsky, 2022; Saxe et al., 2022).

Here, we characterize the dynamics of nonlinear-nonlinear teacher-student scenarios. We examine networks with zero-bias linear layers and ReLU nonlinearities (Fukushima, 1975; Nair & Hinton, 2010; Wang et al., 2016; 2019; Mohan et al., 2019; Keshishian et al., 2020; Srinivas & Fleuret, 2018; Balestriero & Baraniuk, 2020), which allow for exact linear decomposition of the network function via the SVD of the Jacobian for a given high-dimensional input point. By measuring the Jacobian for the locally linear transform at every input point in the dataset, we can apply the techniques used by Lampinen & Ganguli (2018) for measuring the dynamics of generalization to nonlinear student and teacher networks. Below, we investigate a nonlinear-nonlinear teacher-student framework utilizing the SVDs of the collection of Jacobian matrices to examine the mode strength, alignment and loss over training, revealing nonlinear mode coupling, the dynamics of residual nonlinear modes and their manifestations in a real dataset.

## 1.1 THEORY

Lampinen & Ganguli (2018) designed a teacher-student framework to investigate the nonlinear dynamics of learning in deep linear networks. The teacher network is defined by a weight matrix $\boldsymbol{W}_T$ (e.g., $12 * 12$ elements) relating an input vector $\boldsymbol{x}$ (with 12 elements) to an output vector $\boldsymbol{y}$ (with 12 elements). Some number of input vectors is chosen and generated (with whitening, such that the covariance matrix $\boldsymbol{X} * \boldsymbol{X}^T$ is approximately the identity) to create output examples using the weight matrix. The deep linear student network has the form:

$$\boldsymbol{y}_S(\boldsymbol{x}) = \boldsymbol{W}_{S,N} \boldsymbol{W}_{S,N-1} ... \boldsymbol{W}_{S,2} \boldsymbol{W}_{S,1} \boldsymbol{x} \tag{1}$$

where $\boldsymbol{W}_{S,1}$ is the weight matrix of the first student layer.

In order to create train and test sets from the teacher network output, Gaussian noise with a specified standard deviation $\boldsymbol{z}$ that was small in comparison with the singular values of $\boldsymbol{W}_T$ is added to each output vector $\boldsymbol{y}_i$ in the train set. The network is trained to predict the noisy teacher output matrix $\hat{\boldsymbol{y}}_T$ from the input data $\boldsymbol{x}_i$. The deep linear student network has a composite transform that can be decomposed with the SVD:

$$\boldsymbol{W}_S = \boldsymbol{W}_{S,N} \boldsymbol{W}_{S,N-1} ... \boldsymbol{W}_{S,2} \boldsymbol{W}_{S,1} = \boldsymbol{U}_S \boldsymbol{\Sigma}_S \boldsymbol{V}_S^T \tag{2}$$

Here, we consider nonlinear teacher and student networks of the following form with the leaky ReLU nonlinearity as $R$:

$$\boldsymbol{y}_S(\boldsymbol{x}_i) = R(\boldsymbol{W}_{S,N} R(\boldsymbol{W}_{S,N-1} ... R(\boldsymbol{W}_{S,2} R(\boldsymbol{W}_{S,1} \boldsymbol{x}_i)))) \tag{3}$$

The complete network transform is locally linear and fully described by the Jacobian of the network output with respect to an input vector $\boldsymbol{x}_i$:

$$\frac{\partial \boldsymbol{y}_s}{\partial \boldsymbol{x}_i} = \boldsymbol{W}_{S,x_i} = \boldsymbol{U}_{S,x_i} \boldsymbol{\Sigma}_{S,x_i} \boldsymbol{V}_{S,x_i}^T \tag{4}$$

The network's local linear equivalent $\boldsymbol{W}_{S,x_i}$ is a distinct matrix for every input and must be computed numerically with the Jacobian. Other nonlinearities like sigmoid, tanh, CELU, GELU, etc.,

all generate outputs with higher-order terms beyond the Jacobian; ReLU outputs can be completely described with only the Jacobian. Therefore while ReLU student networks can learn arbitrarily complex nonlinear functions, their globally nonlinear behavior can also be interpreted with linear tools like the SVD (Mohan et al., 2019).

The mean-squared error loss that is optimized to train the deep ReLU student network from the teacher can be computed using the mean of the networks' equivalent linear expressions for each example.

$$\frac{1}{k}\sum_{i=0}^{k}(\hat{\boldsymbol{y}}_T - \boldsymbol{y}_S)^2 = \frac{1}{k}\sum_{i=0}^{k}[(\boldsymbol{U}_{T,x_i}\boldsymbol{\Sigma}_{T,x_i}\boldsymbol{V}_{T,x_i}^T\boldsymbol{x}_i + \boldsymbol{z}) - \boldsymbol{U}_{S,x_i}\boldsymbol{\Sigma}_{S,x_i}\boldsymbol{V}_{S,x_i}^T\boldsymbol{x}_i]^2 \qquad (5)$$

Given that the total network operation is linear at the level of individual examples, as each $\boldsymbol{W}_{S,x_i}$ is different, the derivation of singular value strength and alignment over training by Lampinen & Ganguli (2018) holds here for deep ReLU networks. For the $\boldsymbol{z} = 0$ noiseless setting, the student network transform must evolve over training to match the teacher transform; for a given input example, the student singular vectors of the linear transform $\boldsymbol{W}_{S,x_i}$ must align with the teacher singular vectors, and the student singular values must grow to the same magnitude as the teacher singular values.

The magnitude of the student singular values is a vector $\boldsymbol{m}_S$ which corresponds to the average over all examples of each element on the diagonal of $\boldsymbol{\Sigma}_{S,x_i}$. The alignment of the left singular vectors from the student $\boldsymbol{U}_S$ with the teacher $\boldsymbol{U}_T$ was measured by taking the average of the diagonals of each element of their inner product. An equivalent measurement is made for the teacher and student right singular vectors.

$$\boldsymbol{m}_S = \frac{1}{k}\sum_{i=0}^{k}\mathrm{diag}(\boldsymbol{\Sigma}_{S,x_i}) \qquad (6)$$

$$\boldsymbol{a}_S = \frac{1}{k}\sum_{i=0}^{k}\mathrm{diag}(\boldsymbol{U}_{T,x_i}\boldsymbol{U}_{S,x_i}^T) \qquad (7)$$

In the nonlinear-nonlinear teacher-student setting, since the Jacobian $\boldsymbol{W}_{S,x_i}$ differs for each example, the SVD of $\boldsymbol{W}_{S,x_i}$ must also be computed for each training and validation input $\boldsymbol{x}_i$ for both the teacher and student networks. In order to measure alignment, each student singular vector $\boldsymbol{U}_{S,x_i}$ is compared with the teacher singular vector $\boldsymbol{U}_{T,x_i}$. The mean magnitudes and alignments are computed over all examples, over different training runs with random initializations, in the plots shown below in Results. Notably, if this Jacobian method for ReLU networks is applied to deep linear networks, the Jacobian operation will produce the same linear weight matrix for every input point, equal to the deep linear network's composite weight matrix. For deep linear networks, our Jacobian method produces the same results as the method from Saxe et al. (2013) and Lampinen & Ganguli (2018), which reads the student network's composite weight matrix at every training step. For the deep ReLU case, the Jacobian method allows us to compute that equivalent composite weight matrix for every input vector.

## 2 METHODS

### 2.1 EXPERIMENT OVERVIEW

We conducted a number of experiments to examine the dynamics of learning for nonlinear zero-bias deep ReLU teacher and student networks. Inputs and outputs were both 12-dimensional vectors. For the deep linear case with zero noise, the number of examples only needs to be the same as the dimensionality of the input. However, for the deep nonlinear case, the number of examples must be larger in order to accurately learn the transform, and we generally used 256 training examples with 128 validation examples. The weight layers in the deep ReLU networks were 12-dimensional square matrices, matching the dimensionality of the input and output. Gaussian noise was added to the output train examples in the noisy training scenario with varying magnitudes. A learning rate of $1.8e\text{-}5$ was used with the full-batch gradient descent optimizer unless otherwise noted.

## 2.2 Specifying the singular value spectrum in nonlinear teacher networks

In order to create a nonlinear teacher network with a specified singular value spectrum, weights for each layer are randomly initialized as orthogonal square matrices. The singular values of the first layer weight matrix are scaled to the desired magnitudes. For example, an appropriately scaled random orthogonal matrix has most singular values at close to 1; to generate five "signal" singular values, five singular values are scaled to magnitudes $50, 40, 30, 20$ and $10$, while the rest are set to zero, as in Figs. 1A and 1D. A new first layer matrix is then constructed from this rescaled singular value spectrum and the original singular vectors. The subsequent layer weight matrices are not altered from their random orthogonal state.

## 3 Results

### 3.1 Mode decomposition of loss in deep ReLU networks

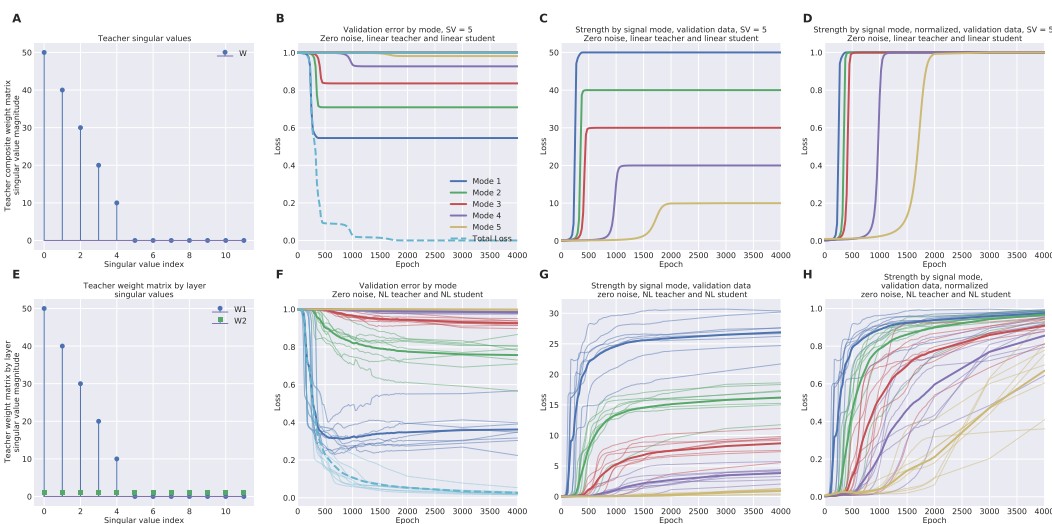

Figure 1: Loss decomposition by singular mode. A) The teacher network's composite singular value spectrum for a deep linear network. B) For a linear teacher network and linear student network, the validation loss of the linear student network over training and the loss decomposed by mode. C) The singular values of the weight matrix of the linear student network over training and D) the normalized singular values over training. Note these match the analytic sigmoid solution found by Saxe et al. (2013) E) The singular value spectra for different layers of the nonlinear teacher network. F) For a nonlinear teacher network and nonlinear student network, the validation loss of the nonlinear student network over training and the loss decomposed by mode (see below for details on mode decomposition of the nonlinear student network). The curves for eight training runs with different initializations are plotted to give a sense for the distribution (thin lines). G) The singular values of the weight matrix of the nonlinear student network over training and H) the normalized singular values over training. In the nonlinear-nonlinear teacher-student setting, while the nonlinear singular values generally increase as in the linear-linear case, they do not have a sigmoid evolution over training like in the linear-linear case. The loss by mode shows that the loss due to the first nonlinear mode actually increases over time after a sharp decrease, as the weaker modes are learned.

Fig. 1B shows the loss curve for a randomly initialized deep linear student network trained on data from a zero-noise linear teacher. The loss is also plotted by mode, and the decreasing levels of loss for the linear student network match the teacher network designed with a stepwise decrease of teacher singular mode strengths.

For a similar experiment with nonlinear teacher and nonlinear student networks, where both teacher and student are zero-bias deep ReLU networks with zero noise, we observe loss over training as in Fig. 1E. The teacher network has two linear layers and a ReLU with leak $\alpha$ set to 0.1, where the first layer has a stepwise singular value structure of five signal modes (and the remaining seven

modes are noise), and the second layer is a random orthogonal matrix with a uniform singular value spectrum, as in Fig. 1D. The exact local transforms for each exemplar in the train and validation sets are found by numerically computing the Jacobian with respect to each input vector. The SVD of each Jacobian has on average five singular modes (with some variance in magnitude) greater than zero, representing the nonlinear teacher signal singular modes, and different singular vectors across exemplars. Here the losses for each singular mode for every exemplar are computed and their sums are plotted Fig. 1E.

The student network has three hidden layers with zero-bias leaky ReLUs and reaches validation error on the order of $5e$-3. This requires far more examples than in the linear case (on the order of the number of trainable parameters). Initially the first "nonlinear" mode accounts for most of the loss decrease, but as the second mode accounts for more of the loss decrease as training continues, the loss from the first mode actually increases as overall loss continues to decreases. This is a form of nonlinear mode coupling, at least given the definition of a nonlinear mode as the average of the locally linear modes for every validation example.

## 3.2 SINGULAR MODE STRENGTH AND ALIGNMENT INCREASE DURING TRAINING

Figs. 1C and 1D show the singular mode strengths for the linear-linear case, which follow the sigmoid curves analytically derived in Saxe et al. (2013), delayed in accordance with the strength of the mode in the teacher network. They are learned in a "singular value detection wave" (Lampinen & Ganguli, 2018) from the strongest teacher mode to the weakest. Figs. 1G and 1H show the mode strength averaged over all examples and runs for each training epoch for the nonlinear-nonlinear scenario. While the progression is much noisier than the linear-linear mode strengths, the nonlinear modes have a qualitatively similar profile complicated by mode competition.

Fig. 2A shows singular mode alignment for the linear-linear teacher-student case, for a scenario with five nonzero singular values specified in the teacher transform. Note that the five signal modes are aligned nearly perfectly at $1.0$ in the course of training, and cannot be clearly distinguished on the plot, while the seven noise modes (dashed curves) remain unaligned below.

Fig. 2B shows mode alignment averaged over all examples and eight training runs for the nonlinear-nonlinear teacher-student scenario with five nonzero singular values specified in the first-layer teacher transform, trained on $512$ input-output pairs and with a validation set of $256$ input-output pairs. The mean alignment over all validation example inputs with the teacher transform increases, which is in line with the theory for deep linear networks. However, it is not nearly as clear in the linear-linear setting of Fig. 2A. A nonlinear student network trained with the Adam optimizer (Kingma & Ba, 2014) is shown in Fig. 2C, where the singular vector alignment is much higher.

Another method for viewing the alignment of the singular vectors is by the alignment for each example after training is complete. This is shown for both the the student validation sets in Fig. 2D, 2E and 2F, where the examples have been sorted by alignment. Each curve represents a different mode. In the linear-linear case of Fig. 2D, the signal modes are all at perfect alignment, while the seven noise modes are unaligned. Fig. 2E shows alignment for the nonlinear-nonlinear teacher-student scenario with gradient descent optimization. The signal modes are clearly more aligned on average than the noise modes, but the alignment is far from the linear-linear case. Fig. 2F shows much stronger alignment for the nonlinear student network trained with Adam optimizer.

## 3.3 NONLINEAR COUPLING BETWEEN SIGNAL AND NOISE MODES

Lampinen & Ganguli (2018) observed that noise modes ought to couple to signal modes in nonlinear networks. In the linear-linear teacher-student scenario, the singular modes of a network are necessarily orthogonal and cannot affect one another. However, the average modes in a deep ReLU network do not have to be orthogonal and may show evidence of coupling. This is reflected at an empirical level for the nonlinear-nonlinear teacher-student scenario with only one signal mode as in Figs. 3A and 3B, and one signal mode with a noisy teacher signal in Figs. 3C and 3D.

The validation loss curves for the nonlinear student, when decomposed by mode as shown in Figs. 3B and 3D, reveal that the student network's first mode (corresponding to the only teacher signal mode) initially plays a large role in decreasing the loss, as expected. However, Fig. 3D shows that once the second mode ("mode 1" green curve, a noise mode) appears in the student network

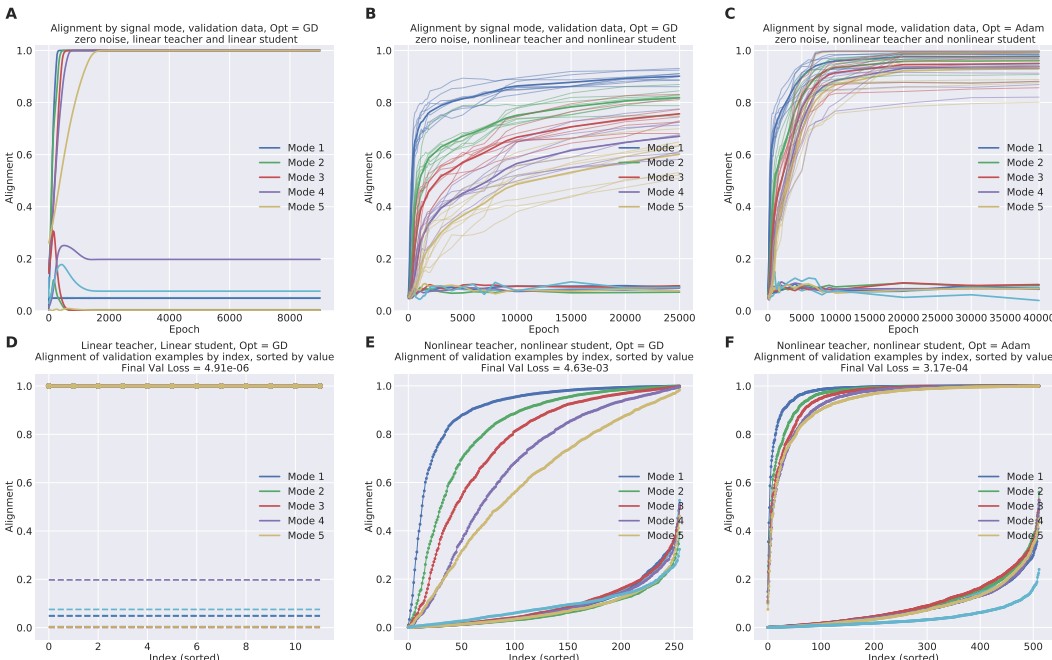

Figure 2: Alignment over training (averaged for all validation examples, eight runs from random initialization) for the experiment described above in Fig. 1. A) The linear-linear teacher-student scenario. Note each of the signal modes reaches perfect alignment with the teacher singular vectors. B) Alignment for the nonlinear-nonlinear teacher-student scenario under full-batch gradient descent optimization. While the mode alignment increases, no mode reaches perfect alignment (due to some validation examples not being predicted successfully, indicative of a failure to completely learn the nonlinear transform) with a final mean validation loss of $4.63e$-3. C) Alignment over time for the nonlinear-nonlinear teacher-student scenario with the Adam optimizer. Mode alignment for each signal mode is much higher in this case than B) and the final mean validation loss is much lower at $3.17e$-4. D) The alignment for each validation exemplar after training for the linear-linear scenario, sorted by increasing value for both signal and noise modes. The signal mode alignment for all five signal modes and every example is perfect at 1, and the curves for the signal modes overlap. The noise mode alignments are lower than 1. The alignment for every exemplar of a mode is the same, because the student transform is linear. E) The alignment sorted in increasing order for validation exemplars in the nonlinear-nonlinear setting with gradient descent. Note that the signal modes have more examples with better alignment, while noise modes for exemplars are always below 0.6. F) With the Adam optimizer, the nonlinear student transform for more validation exemplars is much better aligned than with gradient descent, which is reflected in the lower final validation loss.

transform, the total loss stops decreasing and begins to increase due to learning the noise mode after epoch 20000. Fig. 3H shows the alignment of the first mode begins to decrease on average after this noise mode appears in the student network. The decrease in average alignment of the signal mode shown in Fig. 3H is a manifestation of nonlinear coupling to the noise mode that shows up in the training data for this particular scenario.

## 3.4 THE LINEAR AND NONLINEAR COMPONENTS OF SINGULAR MODES FOR DEEP RELU NETWORKS

Even with a nonlinear deep ReLU teacher network, a linear model can still fit the teacher transform to some degree. A natural next step is to ask how much of the fit by a nonlinear student is accounted for by a least-squares linear fit. Here, we remove the linear component from the teacher train and validation sets, and consider a deep ReLU student network fit to only the residuals of that linear fit.

A least-squares linear model is fit to only the training data, and then the linear prediction is subtracted from each training and validation output exemplar to determine the residual nonlinear component.

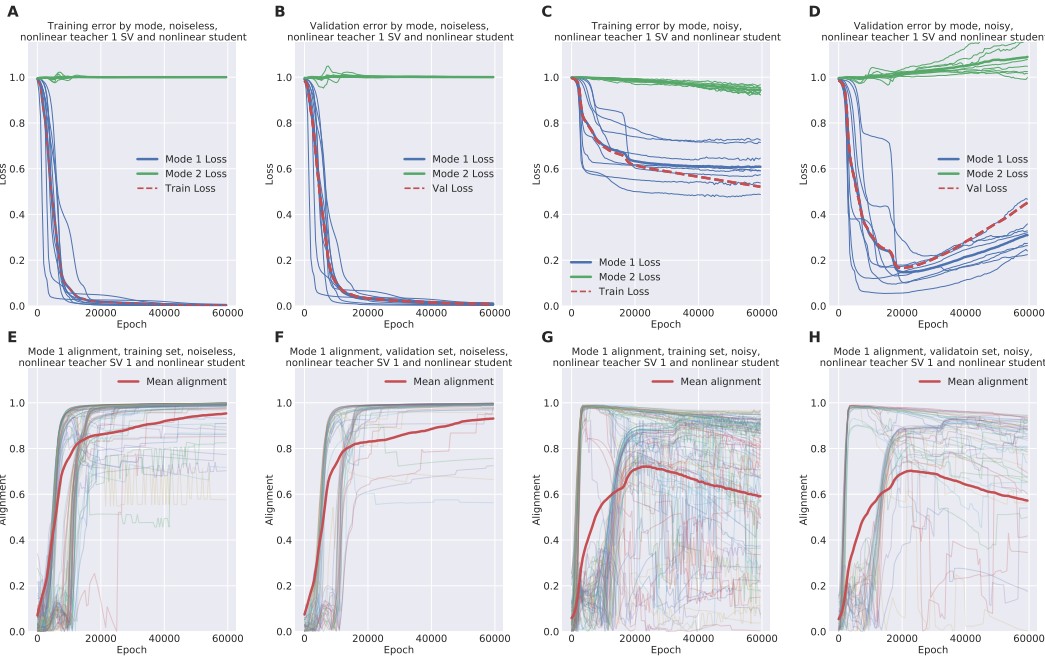

Figure 3: Mode coupling with a noise mode. A) The training loss curve for a nonlinear-nonlinear teacher-student scenario with only one teacher singular vector in the first teacher layer, zero noise and B) validation loss curve, each averaged over eight different training runs from random initializations (thick curve over thin curves of the same color). C) Training loss curve for the same teacher-student scenario from A) with additive white Gaussian noise on the training output and D) validation loss curve for the noiseless output, averaged over eight training runs, allowing for insight into how the student network learns the underlying transform from a noisy signal. The second mode in green in C) is a noise mode that contributes to decreasing the training loss, but in D) that mode increases the validation loss, as the noise is not present in the teacher transform. E) and F) show alignment of the signal mode for each input point for one run and the mean alignment over all runs in red. The initial increase in alignment is for input examples the student fits nonlinearly, while the second increase in alignment at epoch 10000 is for examples fit with the linear component of the nonlinear student network. The alignment of the student singular vectors over training with noise for G) each training example and H) each validation example, as well as their averages over all runs (red curves). Note the time at which the decrease in alignment of the signal singular value occurs slightly after the minimum validation loss near epoch 20000 in D), which is evidence that the noise mode is coupling into the student signal mode and misaligning its singular vectors.

The norm of this residual output signal is then scaled up to match the norm of the original signal. Since the transform now has no linear component but the student network has the same number of parameters, the residual signal is more difficult to learn and takes more training epochs to reach the a similar error.

However, the benefit from an analysis perspective is that the residual nonlinear transform is somewhat simpler in terms of singular mode dynamics than the transform learned from a deep ReLU teacher network. In the previous experiments, each singular mode is actually a composite of linear and nonlinear components, which results in complex dynamics in the mode strengths over the course of training. The loss and singular mode strength for linear-linear teacher-student scenario is shown for one mode in Fig. 4A and five modes in Fig. 4D.

Figs. 4B and 4E show the linear and nonlinear components of the student network over the course of training, where the linear component is measured with a least-squares fit to the predicted output and training input in a given epoch. For the first mode, the nonlinear component accounts for much of the loss immediately, and the linear component is somewhat slower.

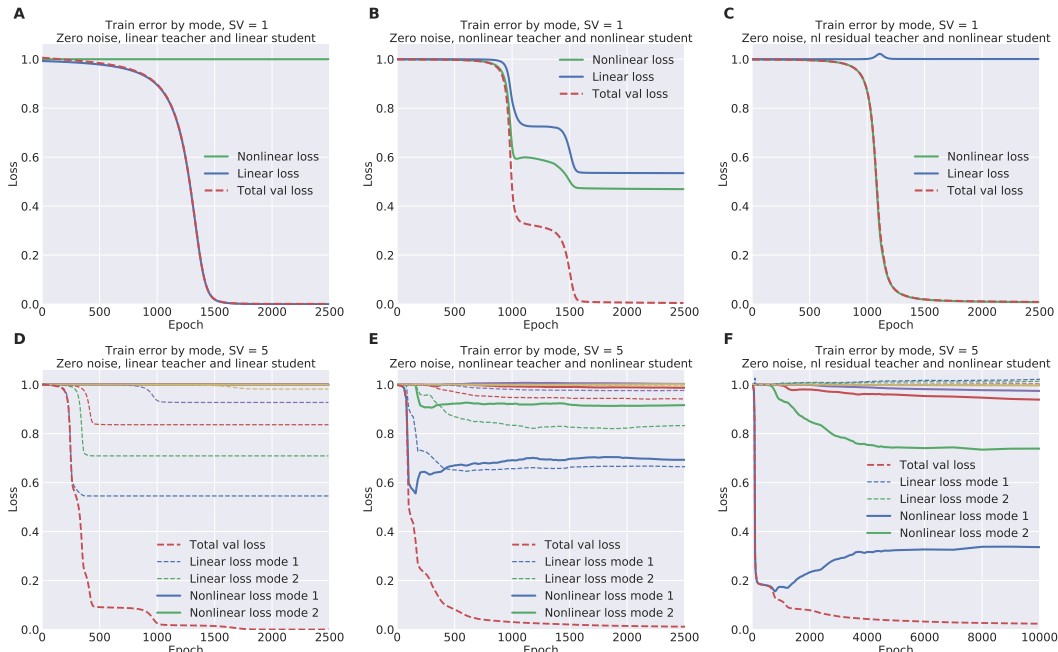

Figure 4: Linear, nonlinear and nonlinear residual teacher settings. Loss curves are for a single run. A) The train loss for a linear student network trained on a linear teacher network with one signal singular mode. B) The train loss for a nonlinear student network trained on a nonlinear teacher network with one singular mode. C) The train loss or a nonlinear student network trained on a nonlinear residual teacher network (where the linear component of the teacher transform has been removed before training) with one singular mode. Note the linear transient that increases the loss (blue curve over 1.0, after epoch 1000). D) The loss for a linear-linear scenario with five signal modes. E) The loss for a nonlinear-nonlinear setting with five modes; there are large linear and nonlinear components of each of the five learned signal modes. F) The loss for the nonlinear residual scenario with five singular modes; note the loss decrease is all nonlinear, but there are small linear components that increase the loss.

Figs. 4C and 4F show the relatively simpler dynamics of the deep ReLU student network trained on the residuals of the deep ReLU teacher network. The modes are learned more slowly due to the difficulty of learning the nonlinear transform to fit the residuals. The removal of the linear component of the teacher signal simplifies the time course of learning for this nonlinear student network. The loss curve of Fig. 4C qualitatively resembles the loss of a ReLU network fit to an XOR dataset in Fig. 3C of Saxe et al. (2022), which suggests that nonlinear residual teacher networks with nonzero singular values may be some form of generalization of the XOR task.

### 3.5    Nonlinear singular modes have interpretive value for models trained on real datasets

Our approach takes advantage of the complexity and transparency of deep ReLU networks to provide insight into the dynamics of networks trained on the synthetic datasets used in the above experiments. Here we demonstrate that the singular mode decomposition of the Jacobian can be applied to a student network trained on a subset of MNIST, and similar training dynamics are observed.

In this case, we trained a three-layer convolutional autoencoder network to cluster images of '0' and '1' from MNIST (LeCun & Cortes, 2010). The size of the autoencoder latent space determines how many singular values of interest there are. For example, with a latent space of five dimensions as examined here, there will be five significant singular values for the Jacobian at each exemplar. The first two dimensions are shown in Fig. 5C). In this case we first train the teacher network on the raw dataset to learn a useful latent representation, and then train a student network with the same structure as the teacher network to directly learn the latent space representation. The strongest mode

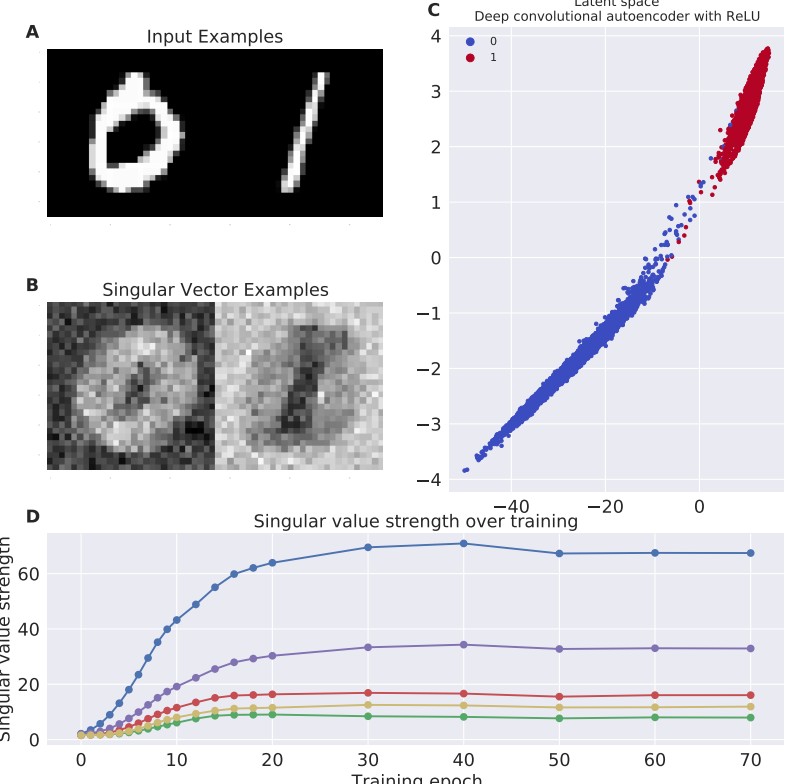

Figure 5: A deep convolutional autoencoder teacher-student with MNIST 0 and 1 examples shown in A). Deep ReLU networks allow the teacher-student scenario to be applied to more realistic datasets. B) Example singular vectors of trained student network for individual examples. C) Latent space representation of test set. D) Singular values over the course of training.

is learned first, followed by weaker modes as in in Fig. 5D). We observe increasing mode strength and alignment with the teacher network over time.

## 4    CONCLUSION

We demonstrate that analysis of the Jacobian of deep ReLU networks provides insight into the dynamics of learning for nonlinear transforms. Shifting the level of analysis to the exact local linear transform employed by the network on average for each exemplar allows for a clear connection to the singular mode decomposition.

We found general trends of increasing singular mode strength and increasing mode alignment with the teacher network when averaged over all training and validation exemplars. We also observed complex mode coupling, both between signal modes and between signal and noise modes, and observed that when a noise mode is learned, it corrupts the alignment of an already-learned singular mode with high average alignment. We probed the linear component of the transform present in deep ReLU teacher networks, and examined learning dynamics for only the nonlinear residual component of the teacher network. Finally, we demonstrated similar trends in the nonlinear student singular modes of a network trained on MNIST data.

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
