# OpenReview forum: "An empirical investigation of generalization dynamics in deep ReLU networks via nonlinear mode decomposition"
_ICLR.cc/2024/Conference — ICLR 2024 Conference Withdrawn Submission_

### Official Review · Reviewer_wkT2 · 2023-10-27

**Soundness:** 2 fair
**Presentation:** 2 fair
**Contribution:** 2 fair
**Rating:** 3
**Confidence:** 3

**Summary:**

The authors consider the singular value decompostion of the Jacobians of networks for the teacher-student senario.
They empirically investigate the behavior of the modes for linear and nonlinear networks.

**Strengths:**

The relationship between the generalization property of neural networks and spectral propseties of Jacobians of the networks has been investigated, and an interesting topic.
The authors show several numerical results related to this topic.

**Weaknesses:**

The experimental setting should be more realistic.
In my understanding, the authors constructed a network with squared weight matrices, whose sizes are 12 by 12.
The number of training samples is only 256.
They also considered a CNN and MNIST.
In practical situations, we often come across larger networks and a larger number of training samples.
If the paper is based on empirical results, the experiments should be conducted in more pratical and various settings, e.g., AlexNet and VGGNet.

**Questions:**

The authors show the reslts of ReLU networks. Does the empircal results for other activation functions differ from those of ReLU?

---

### Official Review · Reviewer_LBAp · 2023-10-29

**Soundness:** 1 poor
**Presentation:** 1 poor
**Contribution:** 1 poor
**Rating:** 3
**Confidence:** 5

**Summary:**

This paper studies zero-bias deep ReLU networks with a teacher-student framework. The paper has provide a series of expriments, most synthetic data and some MNIST dataset. The key technique used in this paper is the singular value/mode decomposition (SVD) of the Jacobian computed at every input point for both teacher and student networks. Authors explore the evolution over training of the singular values and vectors, averaged over all inputs. They show over the course of training increasing student singular value magnitudes and increasing alignments of student singular vectors with teacher singular vectors, as observed in deep linear networks.

**Strengths:**

There are a series of experiments to support the claim.

**Weaknesses:**

1 writing is poor.
2 condition is too restrictive.
3 related works are not surveyed enough.
detailed is in Questions.

**Questions:**

1 the writing are extremely poor. Many concepts are not clearly defined or explained. For examples, a) how to plot loss by mode? b) at the end of page 4, Fig. 1E should be a typo. c) In Fig. 2, there are two sets of curves with the same color, what is the meaning of each curve? d) at the end of page 7, what is the linear and nonlinear components? e) Fig. 5C is not clear explained.

2 why the loss in the Fig. 1B of different modes are larger than the total loss?

3 There are a series of works analyzing spectral of neural network, however, authors seems now aware of such works, such as
Training behavior of deep neural network in frequency domain.
On the spectral bias of deep neural networks.
The convergence rate of neural networks for learned functions of different frequencies.
A fine-grained spectral perspective on neural networks.
On the exact computation of linear frequency principle dynamics and its generalization.
It is really hard to see a novel contribution of this paper.

4 zero-bias a very restrictive condition.

---

### Official Review · Reviewer_dQaM · 2023-10-29

**Soundness:** 3 good
**Presentation:** 2 fair
**Contribution:** 3 good
**Rating:** 6
**Confidence:** 3

**Summary:**

The authors' empirically study the behavior of deep ReLU networks, extending work that focused on deep linear networks. The authors use a mode decomposition technique to study how different modes evolve with training time, how they are coupled, and how this affects the loss behavior of the networks under study. Comparing their results to deep linear networks, they seem similarities (increasing modes increase their strength), but the shape the evolution of the strength of the modes takes, as well as the coupling between modes, is different than deep linear networks. The authors test a prediction made previously in work done on deep linear networks, namely that noise modes and signal modes would couple. Finally, the authors provide evidence that their analysis can hold for networks trained on more standard data sets (e.g., MNIST).

**Strengths:**

1. This paper describes how linear deep neural networks have been deeply studied, but this work has largely been limited in its application to nonlinear deep neural networks. This motivates the problem nicely.

2. The authors do a good job (for the most part) comparing their results to the case of deep linear networks.

3. The authors perform the same analysis on a nonlinear residual task, which I thought was clever.

4. The paper is (for the most part) well written.

**Weaknesses:**

Despite the strengths, there are several major weaknesses of the paper. I have decided to award a score of 6 to the authors, but this is viewed as a temporary score: if the points below are addressed I would be inclined to raise my score; if they are not I will decrease my score.

1. The decomposition of the ReLU network, Eq. 4, which is the key to all the analysis that follows, has no proof or citations. Perhaps it's an obvious conclusion, but it was not entirely clear to me why this would always be true. Discussion on this, or maybe a brief proof in an Appendix, would start the paper off on a stronger footing.

2. Figures 1, 2 and 4 have a number of issues:
    i. It is quoted in the main text that dashed lines are used to denote noise modes but, aside from Fig. 2D, there are no dashed lines (Fig.
       2).
    ii. There are more bolded lines in the plots than there are in the legend (Fig. 2 and 4).
    iii. What "Nonlinear loss" refers to in Fig. 4A, where the network is linear, is not clear.
    iv. The y-axis of Fig. 1C, D, G, H should read strength not loss correct?
These issues make it difficult to interpret the results.

3. I find the increase in alignment of, what I believe are the noise modes - although again, the plots make this challenging to be sure, with index in Fig. 2E, F very interesting. However, this is not commented upon at all in the main text. Discussion is needed on this point.

4. I thought I understood what normalized modes meant (Fig. 1D, H), but the fact that several of the modes in Fig. 1H (the nonlinear network) do not end at 1 makes me think I misunderstood. What exactly is meant by normalized should be discussed in more detail.

5. It is said that "The teacher network has two linear layers and a ReLU with a leak ..." (end of page 4). Does this mean a single ReLU at the end of two linear layers? If so, isn't this different than what is considered in Eq. 3?

**Questions:**

All my questions are stated in detail in the section above.

**Details Of Ethics Concerns:**

I found no ethics concerns in this work.

---

### Official Review · Reviewer_CB8g · 2023-10-30

**Soundness:** 2 fair
**Presentation:** 1 poor
**Contribution:** 2 fair
**Rating:** 3
**Confidence:** 2

**Summary:**

This paper investigates the generalization dynamics of deep ReLU networks in a teacher-student framework. The authors use singular value decomposition (SVD) of the Jacobian to measure the globally nonlinear behavior of the network transform. They find that the singular values and vectors of the student network evolve over training, and that the network's ability to learn from the teacher network is influenced by the number of singular modes greater than zero. The authors conclude that their findings have implications for the development of more effective deep learning algorithms and models. Overall, the paper contributes to our understanding of the generalization dynamics of deep ReLU networks and provides insights into how to improve their performance.

**Strengths:**

The authors use a novel approach of applying singular value/mode decomposition (SVD) to measure the globally nonlinear behavior of the network transform in deep ReLU networks in a teacher-student framework. This is a unique and original approach that has not been extensively explored in previous research.

The authors use singular value decomposition (SVD) to analyze the evolution over training of the singular values and vectors of the Jacobian of the network transform with respect to every input point. They show that for a deep ReLU student network trained on data from a deep ReLU teacher network with a user-specified singular value spectrum, there is an increase in student singular value magnitudes and increasing alignments of student singular vectors with teacher singular vectors, as observed in deep linear networks.

The authors also decompose the loss over training by singular mode and directly observe nonlinear coupling of noise to student signal singular modes as well as coupling due to competition between signal modes.

This paper provides insights into how to improve the performance of deep ReLU networks and represent a valuable contribution to the field of deep learning.

**Weaknesses:**

Insufficient experiments: expanding the experimental evaluations to encompass more diverse datasets and network architectures would better demonstrate the generalizability of the phenomenon. The current experiments focus heavily on MNIST with deep ReLU networks. Evaluating the effects across more tasks and models could further substantiate the claims.

Lack of clarity: there are some sections that could benefit from more detailed explanations. For example, the authors could provide more information on the methodology used to compute the exact local transforms for each exemplar in the train and validation sets. Additionally, the authors could provide more information on the decomposition of the loss over training by singular mode, including how this was computed and what insights it provides.

Presentation: There are also points worth improving in the writing of the article. For example, the conclusion of the article can be appropriately added to the introduction to help readers read the article; and the conclusions in the Results section of the article can also be described in a better way to show the relationship between each subsection.

**Questions:**

In the experiments, you used the MNIST dataset for image classification. Have you considered using other datasets to test the generalization dynamics of deep ReLU networks, such as CIFAR-10 or ImageNet?

In the paper, you mention that the teacher network was trained on the entire MNIST dataset, while the student network was trained on a subset of the dataset. How was the subset chosen, and what was the size of the subset relative to the entire dataset?

The paper discusses the behavior of deep ReLU networks in a teacher-student framework. Have you considered comparing the results to those obtained from other types of networks, such as convolutional neural networks or recurrent neural networks?